# Zinc-Based Nanomaterials for Diagnosis and Management of Plant Diseases: Ecological Safety and Future Prospects

**DOI:** 10.3390/jof6040222

**Published:** 2020-10-13

**Authors:** Anu Kalia, Kamel A. Abd-Elsalam, Kamil Kuca

**Affiliations:** 1Electron Microscopy and Nanoscience Laboratory, Department of Soil Science, College of Agriculture, Punjab Agricultural University, Ludhiana 141004, Punjab, India; 2Agricultural Research Center (ARC), Plant Pathology Research Institute, Giza 12619, Egypt; kamelabdelsalam@gmail.com; 3Department of Chemistry, Faculty of Science, University of Hradec Králové, 500 03 Hradec Králové, Czech Republic

**Keywords:** ecotoxicity, nanomaterial, nanosensors, phytopathogens, zinc

## Abstract

A facet of nanorenaissance in plant pathology hailed the research on the development and application of nanoformulations or nanoproducts for the effective management of phytopathogens deterring the growth and yield of plants and thus the overall crop productivity. Zinc nanomaterials represent a versatile class of nanoproducts and nanoenabled devices as these nanomaterials can be synthesized in quantum amounts through economically affordable processes/approaches. Further, these nanomaterials exhibit potential targeted antimicrobial properties and low to negligible phytotoxicity activities that well-qualify them to be applied directly or in a deviant manner to accomplish significant antibacterial, antimycotic, antiviral, and antitoxigenic activities against diverse phytopathogens causing plant diseases. The photo-catalytic, fluorescent, and electron generating aspects associated with zinc nanomaterials have been utilized for the development of sensor systems (optical and electrochemical biosensors), enabling quick, early, sensitive, and on-field assessment or quantification of the test phytopathogen. However, the proficient use of Zn-derived nanomaterials in the management of plant pathogenic diseases as nanopesticides and on-field sensor system demands that the associated eco- and biosafety concerns should be well discerned and effectively sorted beforehand. Current and possible utilization of zinc-based nanostructures in plant disease diagnosis and management and their safety in the agroecosystem is highlighted.

## 1. Introduction

Microbial pathogenic diseases of crop plants account for substantial annual loss (in the relative manner depicted as a percentage), approximately 16–40%, of production tonnage [1]. The bacterial and fungal pathogens of various crops exhibit enormous yield and productivity losses during production and postharvest storage as well as during transportation of the crop produce [2]. To safeguard the crop from crop health and yield deterring pathogens, pesticides- organic or inorganic compounds, or their composites have been used by agriculturists or farmers. Among the diverse pesticidal agents utilized to curb weeds and plant pathogens, zinc and copper formulations have emerged as the best performers.

### 1.1. Use of Zinc Element as a Pesticide

Zinc alone or in combination with copper has been widely used for the development of several commercially available agricultural bio-/pesticides [3]. In the early 1970s, zinc salts for pesticide use were first registered in the United States. Later in the 1990s, the US Environment Protection Agency (US-EPA) approved three zinc salts, namely, zinc chloride, zinc oxide, and zinc sulfate for use as herbicide and the industrial preservative (to control spoilage by bacterial and fungal contaminants in carpets) [4]. Zinc phosphide, another Zn-salt, is applied as an effective rodenticide [5,6]. Further, zinc oxide has been approved to be used as a stabilizer in pesticide formulations with concentration not exceeding 15% (*w/w* or *w/v*) of the formulation [7]. Later, the zinc formulations have got popularized for the antimicrobial activity against various phytopathogens. The antimicrobial potential of the zinc formulations render its use as a considerably low cost, less environmentally toxic, and effective microbicide exhibiting broad-spectrum activities including bactericide [8], fungicide [9,10,11], or algaecide [12] and other activities. A growing interest exists for the development of novel zinc formulations possessing enhanced efficacy and action specificity. Generation and use of nanoenabled formulations of pesticides are one among the emerging and pertinent alternatives to manage plant diseases causing phytopathogens.

### 1.2. Status of Use of Nanomaterials in Plant Pathology 

Changing climatic patterns and intensive agriculture has contributed enormously to the development of more fastidious and virulent pathogens, which exhibit resistance to several pesticides (bactericides, fungicides, and similar action compounds) [13,14,15]. These strains of microbes can survive through higher concentrations of the -cidal compounds/composites besides requiring multiple applications and therefore, have become a big menace for the farmers to avoid or control the yield losses caused by these pathogens [16]. The use of nanomaterials for control of phytopathogens has been envisioned by agriscientists after the evidence for -static to -cidal properties of various types of nanomaterials that appeared for human/livestock pathogens in journals of repute of biomedicine or pharmacology [17,18,19,20,21]. Amenability to fabrication/alteration of size and surface morphology and functionalization of nanomaterials is of tremendous significance considering the quick and sustainable eradication of pesticide-resistant phytopathogens [22,23,24,25,26,27].

Various categories of nanomaterials have been evaluated for their diverse agriapplications such as nanofertilizers, nanopesticides, and pesticides degradation to achieve plant growth promotion and protection [28] (Figure 1). Thus, the current manuscript entails the published research on the use of zinc nanomaterials for management and early diagnosis of phytopathogens. Further, the application of zinc nanomaterials as potent antimicrobial agents and their use for curbing the growth, virulence, and diseases caused by plant pathogens have been elaborated. The use of zinc nanomaterials as functional elements in biosensor systems for robust and sensitive identification of phytopathogens is also discussed.

## 2. Nanomaterials: Can Nanosizing Matter Alter Its Properties?

Nanomaterials (NMs) exhibit enormous chemical diversity and can be categorized on basis of their chemical origin as natural, organic, synthetic, metal/nonmetal, or their oxides, sulfides, nitrides, and other forms [29]. These are considered as an intermediate state of matter with at least one of the size dimensions existing between size scales of 1–100 nm. The dimensionality classification of NMs segregates these as zero-, one-, two- and three-dimensional materials [30,31]. The nanomaterials exhibit novel physical, chemical, and biological properties [32,33]. The reason for the unusual properties of nanomaterials may be attributed to the basic phenomena of “quantum confinement” and “surface-interface effects” [34,35,36]. These two characteristics may alter the mechanical, optical, electrical, magnetic, and chemical catalysis properties of nanoscale materials compared to their bulk counterparts [37,38]. Thus, nanomaterials exhibit properties that are size dependent, i.e., the size of grain or particles, phase inclusions, pores, or other morphological features affect the properties exhibited by the substance [39].

### 2.1. Mechanism of Antimicrobial Activity

The antimicrobial potential of the nanomaterials gets improved possibly due to enhanced surface of contact with the microbial surfaces or biomolecules [17,40,41]. On interaction with the microbial cells, NMs can adsorb to oppositely charged functional groups [42] and exhibit the advantage of trespassing the intact cell boundaries/membranes. Further, NMs can generate photocatalytic or redox driven electron/hole or electron–hole pair leading to the formation of reactive oxygen moieties (superoxide anion radicals, hydroxyl radicals, singlet ion, and hydrogen peroxide), which can cause random and rapid oxidation of diverse biomolecules of critical structural, functional, and hereditary role in the cell such as proteins, enzymes, lipids, and nucleic acids [25]. Alternatively, NMs may form complexes with the biomolecules leading to damage and inactivation of biomolecules particularly the proteins [27,43]. These interactions and transformations of the biomolecules result in inhibition of cell growth and division [44]. The distortion of the cell morphology and topography is a common feature epitomized by disruption of cellular membrane including exfoliation or erosion of the membrane bilayer structure, appearance of pits due to preferential dissolution of extrinsic proteins, and leakage of cell cytoplasm or even bursting of the cell [17] (Figure 2). Therefore, the complex cascades, diversity, and multiplicity of these interactions may not allow the pathogen to develop the neutralizing or counter-acting mechanisms to address all these interactions. Thus, NM-based antimicrobials will exhibit durable efficacy as there are fewer chances of development of profound resistance in the pathogen [25].

#### 2.1.1. Metal/Metal Oxides, Metalloid, and Nonmetal Nanomaterials

Plants are affected by diverse biotic stress agents, particularly the phytopathogens that cause various diseases and claim the growth and yield losses in crop plants. The incidences of quick emergence of novel pesticide-resistant phytopathogens and reduced efficacy of already available arsenal of antipathogenic compounds/formulations have led towards a possibility of use of antimicrobial potentials of the nanomaterials to curb plant pathogen, which cause diseases culminating to high economic losses due to crop failure. Metal/metal oxide nanoparticles exhibit appreciable antimicrobial activities, which may span over -cidal to static potentials and help in curbing bacteria (bactericide) [17,18], fungi (fungicide) [40], virus (viricidal) [45], and algae (algicidal) [46].

The antimicrobial effect of metal/metal oxide, metalloid, and nonmetal nanomaterials on the test pathogens have been reported to be size and dose dependent [26,47,48]. Further, substantially low concentrations of nanomaterials are required to achieve significantly improved antimicrobial efficacy as compared to the standard reference antimicrobial agent (such as antibiotics and pesticides) [8,49]. Interestingly, the combinatorial use of nanomaterials along with the conventional antimicrobial agents [50] or a combination of metal/metal oxide/nonmetal oxide NPs can enhance the action-spectrum and reduces the minimum inhibitory concentrations (MIC) values [51]. 

Among the various inorganic nanomaterials, the antimicrobial activity including the antimycotic potential of the noble metal nanoparticles (Au/Ag NPs) against plant pathogenic microbes was identified initially [51,52,53,54,55]. Later, nanoparticles/nanomaterials of Group IIa metals including magnesium [56,57]; calcium [58]; other transition metals such as copper [57,59,60,61,62], iron [61], manganese [57], nickel [63,64], titanium [61,65], zinc [56,57,60,62,66,67,68], and zirconium [21,69]; and nonmetals such as silicon [57], selenium [70,71,72,73], and tellurium [74,75] have been evaluated for their antimicrobial potentials. However, chemically, physically, and biologically synthesized noble metal NPs (Au/Ag NPs), copper/copper oxide, zinc/zinc oxide NPs, and magnesium NPs have been mostly reported for the plant pathogenic microbes, whereas the rest of the NP-microbe studies involved evaluation of antimicrobial activity against human or food pathogenic microbial cultures.

##### Mechanism of Antibacterial Activity of Nanomaterials

Nanomaterials exhibit antibacterial potentials manifested as disintegration of the cell membrane leading to leakage of the cytoplasmic contents followed by the lysis of the bacterial cells [47,76,77]. Passive internalization of the NPs can occur through porin-ion channels in Gram-negative bacteria [78], whereas in Gram-positive bacteria, presence of thick cell wall hinders passive internalization and therefore, dissolved ionic species (e.g., Zn^2+^, Cu^2+^, and Fe^2+^ ions) released by the nanoparticles in vicinity of the cell surface get chelated by lipoteichoic acid [79]. Once inside the cell, the internalized NPs may elicit Fenton- or non-Fenton-based ROS-mediated damage of the plasma membrane, internal macromolecules, and other soluble and catalytic biomolecules [78]. Eventually or simultaneous release of ions by the dissolution of NPs leads to metal/nonmetal ion toxicity culminating to cell death [25,76]. Another interesting mechanism involves inhibited expression of the quorum-sensing regulated genes or functions in bacteria leading to inhibition of the biofilm formation [41,80]. The nanostructured materials can also help in the inhibition of the preformed biofilms of the plant pathogens, which is of great significance for the eradication of resistant bacterial pathogens [81] or pathogens related to food spoilage [80,82]. 

##### Mechanism of Antimycotic Activity of Nanomaterials 

Enormous literature on antifungal potential of nanoscale silver [52,53,54,59,83,84], copper/copper oxide [59,62], and zinc/zinc oxide [40] materials exists (Figure 3). The fundamental benefit of the nanoparticulate fungicide is the performance of these formulations equivalent or superior to the respective bulk salt formulations at relatively lower application doses thereby effectively addressing the phyto- and ecotoxicity issues posed due to the release of the metal cations [85]. There exist multiplexed nanomaterial–fungal cell interactions. The nanomaterial internalization in the fungal cell occurs through three possible mechanisms; (i) nonspecific but direct internalization of the small-sized, mostly, spherical nanoparticles, (ii) specific receptor-mediated adsorption followed by internalization of the NPs, or (iii) internalization of dissolved metal/nonmetal ions through membrane-spanning ion transport proteins (Figure 3). Nanomaterials particularly the metal/metal oxide and nonmetal oxide nanoparticles can curb fungal growth through mechanisms that can be dichotomized as (a) antimycotic effect due to generation of ROS and dissolution of the nanoparticles in the cell environment to release specific ions leading to metal/nonmetal ion toxicity and (b) regulation of the mycotoxin-producing genes for decreased or no expression. A detailed illustration of the same for zinc nanomaterials will be incorporated in Section 3.

##### Mechanism of Antiviral Activity of Nanomaterials

The M/MO/NM/NMO nanomaterials possess antiviral activity against microbial [86], animal [87,88,89,90], and human viruses [91,92,93,94,95,96,97] as depicted in several published reports. The green synthesized (microbial/plant cell extract-derived) nanoparticles particularly silver [98] and gold nanoparticles [99] or their composites [98] have been documented to exhibit virus-neutralizing or -inhibiting properties. Likewise, the role of zinc nanomaterials for the virostatic effect [100], virus neutralization, and for immune-modulatory significance against the emerging COVID-19 causative agent [101] has been well identified.

The application of nanomaterials for the control and treatment of viral disorders in crop plants has also been evaluated and established through molecular biology and in planta assays [45]. One-week preapplication of silver NPs at low concentration (50 ppm) on tomato plants decreased the disease severity and induced systemic resistance against two common tomato viruses, namely, Tomato mosaic virus, and Potato virus Y [102]. However, another in planta study showed significant inhibition of Tomato spotted wilt virus on foliar spray of AgNPs (200 ppm) 1 day after artificial inoculation of the TSWV, whereas the lowest and substantially low inhibition was recorded when AgNPs were applied along with and before the virus inoculation, respectively [103]. Similar results have been documented by Elbeshehy et al. [104] on foliar spray treatment of biogenically synthesized AgNPs derived from cell-free extracts of three *Bacillus* bacteria species (*B. pumilus, B. persicus*, and *B. licheniformis*). Complete inhibition of typical disease symptoms was recorded when the AgNPs were applied (concentration: 0.1 µg µL^−1^) 24 h postinoculation with bean yellow mosaic virus in fava bean cv. Giza 3 variety, whereas weak symptoms were recorded when AgNPs formulation was sprayed on foliage simultaneously to that of swab inoculation of the fava bean plants. Low concentration of fungus generated AgNPs formulation (derived from *Curvularia lunata* cell extracts, concentration: 100 ppm) on spray treatment on the foliage of approximately 1 month (35 days) old tobacco plants (*Nicotiana tabacum* cv. *Xanthi* nc) followed by mechanical inoculation of two leaves (5th and 6th true leaf) with PVY-Ros1 virus after 2 days resulted in 2.67-fold decrease in the appearance of characteristic red lesions/infection loci in AgNP-treated plants. Development of nano-Ag composites can further improve the antiviral activity, for instance, graphene oxide-AgNP composite treatment (at 1 µg mL^−1^) reduced the visible symptoms of disease caused by Tomato bushy stunt virus in test lettuce plants [105]. 

Apart from silver NPs, daily foliar spraying treatment for approximately 2 weeks (12 days) of micronutrient iron oxide NPs (Fe_3_O_4_ NPs, size: 20 nm, concentration: 100 µg mL^−1^) enhanced the resistance of tobacco plants against Tobacco mosaic virus [106]. Another report involved daily foliar spray treatment on *Nicotiana benthamiana* plants with Fe_2_O_3_ (concentration: 50 mg L^−1^) or TiO_2_ NPs (concentration: 200 mg L^−1^) (amount: 5 mL) for 21 days. When these plants were challenged with Turnip mosaic virus (green fluorescent protein-tagged TuMV), the plants exhibited significant inhibition in the proliferation of the inoculated TuMV, particularly decrease in coat protein content as identified through a decrease in the fluorescent intensity of GFP marker in new emerging leaves [107].

## 3. Zinc Nanomaterials and Their Use for Curbing Plant Disease-Causing Pathogens

Metal oxides exhibit substantially high antimicrobial activities. However, the eco- and cytotoxicity aspects associated with the application of these novel antimicrobial formulations have hampered their quick commercial applications. Among the various metal oxides, ZnO nanoparticles appear to be one of the most propitious candidates as these NPs can be generated through low-cost synthesis techniques in bulk amounts. Further, their better biosafety and lower cytotoxicity indices for mammalian cells have been proven through several cell line studies [108,109,110] including the report on the preferential killing of human cancer cells compared to normal cells by ZnO NPs [109]. The antimicrobial action spectrum of Zn nanomaterials includes antibacterial, antifungal, and antiviral characteristics [111]. Therefore, the research insights on relative multifunctional properties of the zinc nanomaterials exhibiting antimicrobial actions are based on a fundamental hypothesis of spontaneous generation of ROS species and intracellular oxidative stress leading to killing of the microbial cells [79,112].

### 3.1. Antibacterial and Mollicute Controlling Potential

The studies involving zinc nanomaterial-antibacterial assay against plant pathogenic bacteria are scarcely reported as the majority published research includes the antibacterial activity against pathogenic bacterial genera/species causing human or animal health diseases [113,114,115]. However, plant pathogenic bacteria-Zn nanomaterial interactions have been studied including the reports showcasing the inhibitory effect on the causative agent of citrus canker (*Xanthomonas citri* subsp. *citri*) [116], rice leaf blight pathogen (*Xanthomonas oryzae* pv. *oryzae*) [81], tomato bacterial spot pathogen (copper-tolerant strains of *Xanthomonas perforans*) [117], the causative agent of lentil bacterial leaf spot (*Xanthomonas axonopodis* pv. *phaseoli*) [118], the causative agent of bacterial blight of lentil (*Pseudomonas syringae* pv. *syringae*) [118], and eggplant bacterial wilt pathogen (*Ralstonia solanacearum*) [119]. 

On the evaluation of the relative antibacterial potential of the Zn-nanomaterials, studies established higher efficacy in comparison to the absolute or conventional bulk controls. Among the green synthesized ZnO NPs derived from three different plant extracts, *Olea europaea* extract-derived ZnO NPs exhibited the highest inhibition zone (2.2 cm at 16.0 mg mL^−1^) for *Xanthomonas oryzae* pv. *oryzae* [81]. Likewise, Graham et al. [108] have compared the relative efficacy of nano-ZnO formulations, Zinkicide SG4 and SG6, in an in vitro assay and showed twofold and eightfold lower MIC for SG4 and SG6, respectively, against *X. alfalfae* subsp. *citrumelonis*.

The antibiofilm forming potential of nanozinc material is of remarkable significance for commercial application. The specific benefit of the antibiofilm property of the zinc nanomaterials [82] spans over the decontamination of the food articles [82], surfaces [120,121], produce processing equipment [122], and packaging systems [80,123,124,125].

Apart from the bacterial pathogens, the crop plants are also affected by obligate parasitic, axenically unculturable prokaryotic cell wall lacking eubacterial plant pathogens [126], the “phytoplasma” or “mollicutes” [127], which are associated with >600 plant diseases across the globe [128,129,130,131]. These initially classified as wall-less bacteria possess a trilaminated unit membrane, a small genome (~680 to 1600 kb), exhibit morphological pleomorphism (size ranging between 0.2 and 0.8 µm, and shapes varying from helical, filamentous, beaded, or simply spheroid), dwell in sieve tubes [132] and therefore, are mainly transmitted by phloem sap-feeding or sucking pest vectors, particularly planthoppers and psyllids, and by vegetatively propagated grafts or tissues [133,134]. Being obligate parasites, phytoplasma diseases can be effectively controlled by managing the vector pest population. Therefore, research efforts to develop RNAi- or dsRNA-based nanoenabled pesticides have been initiated that can effectively control the psyllids and/or leafhopper population [135,136]. However, a few reports have appeared including the development and use of nanoemulsion formulations of antibiotics [137], essential oil or aldehyde compounds (such as cinnamaldehyde), and silver nanoparticles [138] for management or eradication of *Candidatus liberibacter asiaticus* causing Huanglongbing or citrus greening disease. Foliar spray and trunk injection treatments of zinc oxide and zinc sulfide nanoparticles alone as an isopropanol-based emulsion or in combination with cinnamaldehyde-isopropanol have been reported to effectively decrease the occurrence of this bacteria in the phloem tissue [139]. Likewise, published reports indicated in planta inhibition of *Candidatus liberibacter asiaticus* by trunk injection application of aqueous formulations of 4 nm-sized zinc oxide nanoparticles and ZnONP-2S albumin protein composite [140]. A qPCR assay revealed that 1:1 proportion of ZnONPs: 2S albumin (concentration of 330 ppm each) most effectively decreased the bacterial pathogen to about 97% of the initial concentration.

### 3.2. Antimycotic and Mycotoxin Neutralizing/Inhibiting Activity

The antimycotic potential of zinc oxide nanoparticles or its composites has been well identified against phytopathogenic fungi belonging to diverse taxonomic groups/classes such as zygomycetous oomycetes genera (*Peronospora tabacina* [141], *Pythium ultimum, Pythium aphanidermatum* [142]), ascomyceteous genera (*Alternaria alternata* [59,62], *Aspergillus flavus*/*A. fumigatus* [51], *Aspergillus niger* [143], *Botrytis cinerea* [61,62,144,145], *Colletotrichum gloeosporioides* [56,59], *Fusarium graminearum* [146], *Fusarium moniliforme* [40], *Fusarium oxysporum* [66,144,147], *Penicillium expansum* [50,66,144,148]), and basidiomycetous genera (*Erythricium salmonicolor* [68]). 

Zinc-derived nanomaterials (nanoparticles/composites) at substantially low working concentrations can kill spores or exhibit inhibition of spore germination (sporostatic/sporicidal activities) besides inhibiting the vegetative mycelial growth of the filamentous fungal plant pathogens, e.g., a significant decrease in fungal growth of *B. cinerea* and *P. expansum* has been observed on ZnO NPs (3 mM L^−1^ concentration) treatment [144]. Likewise, events of spore germination of *Peronospora tabacina* were observed to be completely inhibited on treatment with Zn NPs, ZnO NPs, and ZnCl_2_ soluble salt at concentrations ranging from 15–20 mg L^−1^ [141].

#### 3.2.1. Mechanism of Antimycotic Activity

Multifarious mechanisms govern the antimycotic activity of the zinc nanomaterials. The primary inhibitory symptoms that appear postincubation of an alive culture of fungi with nanoscale zinc/zinc oxide material include adsorption of nanozinc on the hyphal cell surface, hyphal deformation leading to morphological alterations in the cell wall and cell membrane, formation of sunken or swollen mycelia besides extensive thinning, and branching of the mycelia [144]. The same could be or may not be accompanied by suppression of spore or conidia-forming structures or formation of distorted sporangiophore/conidiophore and absence of formation of perennation structures (spores/conidia) or their number is decreased. Fungal spore nanozinc incubation studies have revealed a delay in spore germination, formation of abnormal stout/short germination tubes, or complete inhibition of spore germination indicating the sporistatic to sporicidal properties of nanoscale Zn material [141]. 

At the cell ultrastructural level, changes in the cell wall and membrane structure epitomized as enhanced thickening of the cell wall, liquefaction of cell membrane, dissolution or disorganization of the cytoplasmic organelles, hypervacuolization, and detachment of cell wall from cytoplasmic contents indicating incipient plasmolysis like features appear [68]. 

At the molecular biology scale, the nanoscale Zn materials exhibit interactions with a variety of biomolecules leading to complexation with structural and soluble proteins, inactivation of catalytic proteins, ROS-mediated damage to nucleic acid, particularly the scission of DNA strand, and breakage leading to chromosomal aberrations [25,143,149]. Interaction of zinc nanomaterials with the hyphal cell surfaces also specifically elicit synthesis of nucleic acid and/or production/secretion of the carbohydrates as depicted through increased Raman spectra signal intensities corresponding to these biomolecules [144]. The production of these components may indicate the increased expression of genes involved in subduing the ROS damage induced by the nanozinc material, particularly the osmolytes such as trehalose oligosaccharide. Further, the cell growth cycle also gets altered thereby inhibiting cell division. 

#### 3.2.2. Mycotoxin Neutralizing/Inhibiting Activity

The effect of nanoscale zinc materials for mycotoxin production by the filamentous fungal hyphae have also been evaluated [150,151]. Mycotoxins exhibit enormous structural and chemical diversity. Several fungal genera produce different types of mycotoxins primarily including aflatoxins (B1, B2, G1, G2, and M10), ochratoxins, deoxynivalenol, trichothecenes produced by ascomycetous genera *Aspergillus* (sexual stage name: *Eurotium*) [152]. Likewise, various species of another ascomycetous fungus, *Penicillium* (sexual stage name: *Eupenicillium*), synthesizes and secretes a variety of secondary molecules considered as mycotoxins such as penicillic acid, brevianamide A, griseofulvin, patulin, citreoviridin, citrinin, roquefortine, cyclopiazonic acid, PR-toxin, fumitremorgin B, penitrem A, luteoskyrin, ochratoxin A, rugulosin, verrucosidin, verruculogen, viridicarumtoxin, and xanthomegnin [153,154]. Ascomycetous member belonging to order Hypocreales, *Fusarium*, produces trichothecenes (including fumonisins, zearalenone, deoxynivalenol, and diacetoxyscirpenol) besides fusaproliferin, beauvericin, enniatins, and moniliformin [155]. Alkaloids of *Claviceps* sp. are also considered mycotoxins and include clavines, lysergic acids and their amides, and ergopeptides [156,157,158]. Besides these genera, *Alternaria* sp. produces diverse types of mycotoxins such as altenuene, alternariol, and its methyl ether, altertoxin, and tenuazonic acid [159].

The engineered NPs including ZnO NPs can control mycotoxin production by the mycotoxigenic fungi besides neutralization or adsorption of already formed/secreted mycotoxins [160] (Figure 4). The antimycotic potential of the nano-Zn formulations has already been discussed in Section 3.2.1. The other two mechanisms that are directly responsible for alteration in mycotoxin production by the mycotoxigenic fungi on supplementation of nanozinc formulations in culture/growth media will be dealt with here. Metal oxide nanoparticles exhibit classical size quantization effect such that discrete energy state appears and the number of surface atoms to bulk ratio gets altered besides the changes in the surface topology, which result in enhancing the reactive surface area [161]. Likewise, the thermodynamics of chemical reactivity is varied due to variations in the surface free energy of the NPs. These features adorn NPs the excellent adsorption characteristics. Though classically, carbon nanomaterials, including the amorphous carbon, graphene oxide, carbon nanotubes, and carbon fullerol nanoparticles [150], carbon nanocomposites [162], and inorganic nanocomposites such as MgO-SiO_2_ nanocomposite [163] and organo-silicate composites [164], exhibit higher potential for mycotoxin adsorption. However, a recent study on the application of fullerol nanoparticles (FNP) on the aflatoxin biosynthetic pathway in *Aspergillus flavus* has been performed which suggested a concentration-dependent eliciting effect of FNP on aflatoxin synthesis after 120 h of incubation [165]. Therefore, other nanoadsorbent alternatives including the metal and metal oxides particularly the iron, copper, silver, and the zinc NPs [150,166] can be evaluated for mycotoxin adsorption and removal. A research study on flower-shaped zinc nanostructures (Znstr) revealed that supplementation of low concentrations of Znstr (1.25, 2.5, and 5.0 mM) in the liquid growth media led to substantial suppression (97%) of aflatoxin biosynthesis by *Aspergillus flavus* besides reducing the content of aflatoxin (69%) in maize grains [167].

Apart from the use of nanomaterials for adsorption of mycotoxins, a recent study deciphering the mycotoxin inhibition mechanism of the AgNPs reported a fungus-growth-independent decrease in the aflatoxin B1 production in *Aspergillus parasiticus* [160,168,169]. Unlike the above study, a report documented inhibition of both growth and mycotoxin production potential of *Fusarium graminearum* on the application of biogenic zinc oxide nanoparticles [170]. However, Savi et al. [168] have reported appreciable antifungal and antimycotoxigenic potential of various zinc compounds against *Fusarium graminearum*, *Aspergillus flavus*, and *Penicillium citrinum*. Therefore, zinc nanomaterials have great potential for curbing the growth and mycotoxin contamination of food and feed material [171].

#### 3.2.3. Zinc Nanomaterials for Curbing Plant Viruses/Viroid Diseases

Viruses and viroids cause diverse diseases in crop plants on infection and are responsible for enormous losses posing a great threat to crop productivity and food security. Further, there is a lack of an effective plant viral disease control strategy besides the occurrence of a few commercial antiviral formulations, which enhance the threat for effective control of plant viral diseases. The use of nanomaterials for curbing the spread and disease severity of plant viruses is rather in its incipient stage and research reports on the use of silver [102,103], silver-graphene composite [105], iron oxide [172], and Fe_3_O_4_ [106] nanomaterials have been published. However, there is one recent report on the application of zinc oxide nanoparticles on the plant foliage to curb Tobacco mosaic virus infection in *Nicotiana benthamiana* [45]. The details regarding the antimicrobial potential of various zinc nanomaterials against plant pathogens have been summarized and presented in Table 1. 

## 4. Zinc Nanoformulations: In Planta Studies and Crop Plant Responses to Pathogen Attacks

Zinc nanoformulations have been evaluated to curb phytopathogenic infections in various crop plants. The major test crop plants that have been utilized as models to evaluate the antimicrobial potential of the nanozinc products include tomato [67], tobacco [141], pepper [145], rice, and wheat [174]. The antibacterial potential of ZnO NPs against *Pseudomonas syringae* pv. tomato DC3000 that causes bacterial speck disease in tomato [67] has been reported. In planta greenhouse study performed with *Lycopersicon esculentum* cv. Pantelosa transplants involved foliar spray treatment of ZnO NPs (100 μg mL^−1^) at a five-leaf stage, which significantly reduced the disease severity as compared to untreated control post-1 week of inoculation of the bacterial pathogen. Further, the researchers also indicated elicitation of the plant’s innate defense system through physiological and biochemical studies including antioxidant enzyme activities and profound vegetative growth [67]. Another interesting study involving the effect of ZnO NPs on synthesis and secretion of signal compounds (siderophores-pyoverdine) by plant growth-promoting rhizobacteria-*Pseudomonas chlororaphis* O6 improved the lateral root formation in wheat plants besides enhancing the immunity of the treated plants [174]. The use of ZnO quantum dots (QDs) surface-functionalized with kasugamycin antibiotic has been evaluated for on-demand pH-responsive release of the loaded antibiotic in a greenhouse study to effectively control *Acidovorax citrulli* and alleviate the disease severity symptoms of bacterial fruit blotch in watermelon seedlings [175]. 

The mixed formulation developed as zinc/copper nanocomposites have also been evaluated for their antimicrobial efficacy under field conditions. Suppression of disease symptoms caused by the Citrus canker causative agent, *Xanthomonas citri* subsp. citri were investigated under field conditions on the application of a ZnO-nanoCu-loaded silica gel (ZnO-nCuSiO_2_ composite) nanocomposite. Young et al. [176] investigated the ZnO-nCuSi for controlling citrus canker disease under field conditions and found that this was effective in suppressing disease at less than half the metallic rate of the commercial cuprous oxide/zinc oxide pesticide, and no phytotoxicity was observed. 

Antifungal activities of ZnO NPs biosynthesized from leaf extracts of *Olea europaea* and *Origanum majorana* plants were evaluated. These NPs significantly reduced the appearance of gray and black mold disease symptoms on artificial inoculation with *Botrytis cinerea* and *Alternaria alternata* in test pepper plants compared to chemically synthesized ZnONPs and untreated control plants [145]. Likewise, a comparative in vivo efficacy study for suppression of *Botrytis cinerea* causing gray mold disease on plum fruits (*Prunus domestica*) by treatment with Ag, Cu, and ZnO NPs at two different concentrations (100 and 1000 μg mL^−1^) was performed [59]. The researchers observed complete inhibition of disease symptoms by AgNPs only while ZnO and CuNPs could help control disease symptoms numerically higher or equivalent to copper hydroxide treatment. A simulation study conducted by Wagner et al. [136] on tobacco leaves revealed the high antifungal potential of Zn nanomaterial against *Peronospora tabacina* primarily through inhibition of the spore germination process. An interactive protective effect of nano-ZnO particle seedling spray/seed soaking followed by seedling spray treatments along with the biocontrol agent, *Trichoderma harzianum,* improved plant’s resistance against the causative agent of damping-off disease (*Rhizoctonia solani*) in sunflower seedlings [177].

Zinc nanomaterials also possess elaborate antiviral properties though the reports on in planta studies involving management of the plant viral diseases are recent and incipient. Hence, little literature is available on this aspect. An in vivo experiment on *Nicotiana benthamiana* involved marked inhibition of replication of the Tobacco mosaic virus on foliar spray treatment of ZnO NPs for approximately 2 weeks (12 days). The replication inhibition process may be attributed to improved growth and induction of plant defense responses as indicated by an escalation in accumulation of ROS, and activity of the ROS mitigating enzyme besides upregulation of pathogenesis resistance-related genes [45]. 

## 5. Zinc-Derived Nanomaterials for the Development of Tools/Devices for Plant Disease Diagnosis

Pathogenic disorders or diseases in plants can be identified through various imaging, spectroscopy, and conjugate imaging and spectroscopy techniques [178]. Most likely, the role of diagnostic techniques is to achieve quick, early, sensitive, simple, in situ, reliable, and automated high throughput identification and quantification of the causative agent so that the extent of virulence can be obtained before the appearance of the actual visual symptoms of the disease [179]. Nanomaterial-based sensor technologies provide flexible and diverse sensing platforms or methods for elucidation/quantification of the single or multiple analytes [180] and can help ensure early, rapid, and sensitive identification of the plant pathogen [181].

The plant produces a myriad of signal molecules in response to a pathogen attack. Few abundant and signature signal molecules including specific enzymes, gaseous molecules (e.g., nitrous oxide, volatile organic compounds), reactive oxygen species, secretory compounds such as oxylipins and expression of a crucial gene (pathogenesis-related proteins-PRPs, PAMPs) can be aptly utilized as key biomarkers for the development of nanobiosensor platforms [178]. As discussed in Section 3.2.2., several mycotoxigenic fungi produce diffusible exotoxins, which can also be utilized as markers for the identification and confirmation of phytopathogenic fungi. A nano-ZnO film-indium-tin oxide electrochemical impedance sensor was developed by coimmobilization of antibodies and BSA protein to detect ochratoxin-A in produce and other plant-derived products [182]. Likewise, DNA aptamer-functionalized ZnO/ZnS quantum dots can help in easy detection of plant pathogens [181].

Sensors systems based on zinc nanomaterials primarily include the semiconductor quantum dot (core–shell, CdSe/CdTe core ZnS shell QDs, and ZnTe or ZnSe QDs)-enabled optical (fluorescence-based) sensors [183]. High luminescence QDs are fascinating nanomaterials that can be used to develop protein–protein/protein–ligand detection assays including the fluorescence resonance energy transfer technique [184]. In fixed cell systems, the QDs can be extensively used as immunohistochemical labels [185]. 

The protein—antibody immunofluorescence-based biosensors are finding sensing applications for plant virus pathogens [186]. Medintz et al. [187] have developed a CdTe/ZnS core-shell QD-based sensor by labelling NeutrAvidin on the surface of biotinylated Cowpea mosaic virus (CPMV) and avidin-decorated QDs, which interacted through the biotin-avidin groups. Further, CdTe/ZnSe core–shell QDs can also be utilized for easy detection of DNA sequence change mutation events [188]. The nano-Zn-based QDs exhibit low cytotoxicity and produce high-intensity fluorescence signals, which have resolutions far beyond the diffraction limit of light [183]. Therefore, these can also be utilized for in planta or in vivo assays. Early and sensitive detection (detection limit of 25 μg mL^−1^) of plant pathogenic *Fusarium oxysporum* has been reported through the use of 3-Mercaptopropionic acid-functionalized CdSe/ZnS QD in a fluorescence-based assay [189]. 

Other than fluorescence-based sensors, nano-Zn enabled optical biosensors have also been developed. One of the most promising applications of these nano-Zn-enabled optical biosensors is quick and sensitive detection of plant pathogenic viruses. A sensitive immune-optical biosensor was developed, which involved immobilization of antibodies against Grapevine virus A-type (GVA) antigenic proteins on a ZnO thin film prepared by the atomic layer deposition technique [190].

As zinc nanomaterials exhibit electron-hole generation due to their semiconductor behavior; these have also been used to develop another category of a sensor system, the electrochemical sensors. Zinc oxide nanorod cyclic voltammetry-based electrochemical sensor has been developed as a disposable sensor for a rapid, cost-effective, and label-free detection of *E. coli* in food matrices [191]. Tahir et al. [192] have investigated the potential of a ZnO-nanocomposite prepared by decorating zinc nanoparticles (25–500 nm) on the surface of multiwall carbon nanotubes to immobilize probe DNA strands having complementarity to Chili leaf curl virus beta satellite. They have assessed the electrochemical performance of this DNA biosensor through the binding of the DNA by cyclic and differential pulse voltammetry scans. A similar kind of electrochemical DNA biosensor has been reported to be developed involving ZnO nanoparticles-chitosan membrane-doped gold electrode to conveniently identify *Trichoderma harzianum* biocontrol fungus [193].

## 6. Potential Application of Zn-Based Nanomaterials and Future Use

Zinc nanomaterials have found elaborate applications in diverse fields of agrirelevance such as for fertilizer nutrient delivery [194] through foliar application [195] or sustained release of nutrient from a nanodelivery vehicle [196,197], as novel antimicrobial agent [40,198,199], pesticide [200], and for environmental remediation [201,202,203]. The role of zinc nanomaterials in nanodiagnostics has been already dealt with in Section 5. Several reports delineating the role of zinc nanomaterials for elicitation of the systemic acquired immunity (SAR) in plants to combat and curb attack by various phytopathogens have been indicating towards the gross positive impact of their use in plant crops [204]. The specific aspects that need to be delved on regarding the voluminous and wide-spread usage of zinc nanomaterials for management of phytopathogens include the development of stable nanozinc formulations and their environmental impacts in the soil food-web on nontarget organisms.

### 6.1. Ecosafety Issues of Nanozinc-Derived Products and Devices

Agriculture is a pivotal global enterprise thrusting the economies of most of nations. Therefore, the products or chemicals utilized for improving the nutritional status (fertilizers) [194,196,197] and for management of the plant pathogenic infections (pesticides) are anticipated to be utilized in quantum amounts. Therefore, a cautious and critical approach is desirable considering the atypical behavior in open, dynamic, and complex multicomponent systems. Further, the ecological nanotoxicity concerns of these materials need to be identified before approving the use of zinc nanomaterial-based agriproducts [205,206,207]. 

A pride and prejudice dilemma exists as zinc nanomaterials, particularly the ZnO NPs, are being exponentially synthesized due to amenability for easy and low-cost production processes [208,209]. Further, the functional versatility of nanomaterials renders them affordable for applications or use in diverse fields spanning over electronics, biomedicine [17,18], environment remediation [210], catalysis [211], agriculture [40], and cosmetics industries. However, the release of Zn nanomaterial through municipal wastewater/sewage water, industrial effluent, and surface wash water drifts these nanomaterials to contaminate diverse soil and water ecosystems posing gradual and subtle to drastic effects on soil and aquatic biota thereby exacerbating the health and sanctity of the contaminated eco-niches [212]. The semiconductor (oxidative stress-inducing) properties and heavy metal nature of the zinc nanomaterials (bioaccumulation) further complicate their ecotoxicity concerns [213] besides the fundamental nanoscale aspects (quantum size effects-size, shape, surface charge-dependent properties, and agglomeration/complexation processes), which lead to diverse cyto-/genotoxic and onco-/mutagenic effects [214]. The nano-Zn material and their dissolution product, i.e., Zn^2+^ ions exhibit toxicity to all types of organisms or biotic components of all trophic levels [215]. Further, the occurrence of other pollutants may enhance the pernicious effects of nano-Zn-based products [214]. Therefore, long-term field studies need to be designed besides improvement in the in silico simulation modeling studies to well predict the aftermaths of the rampant use of nano-Zn-based products in agriculture.

### 6.2. Improved Nanozinc Formulations: The Scar and Sanctity of Stability and Biosafety

Zinc nanomaterials can be synthesized using physical and chemical techniques [216]. However, several reports have considered the biologically synthesized nanozinc formulations to be cost-effective, ecosafe, and stable even under ambient storage conditions [217]. Further, higher antimicrobial efficacy and improved photocatalytic activity were reported for the zinc oxide nanoparticles synthesized from the neem leaf extracts [217]. Although the researchers reported a slight difference in the mean size of the ZnO NPs (sol–gel: 33.20 nm and biosynthesized: 25.97 nm), they have argued that the improved efficacy of the neem extract-derived ZnO NPs was due to greater stability of the dispersion owing to surface functionalization by the leaf phenolics or terpenoids. 

The stability of nanozinc formulations is governed by size-dependent phenomena. Further, the zeta potential and the surface charge ensure the aggregation, flocculation, or sedimentation of the nanoparticles [218]. Most likely, the zinc nanoformulations are made stable by altering either the charge (charge-stabilized dispersions) or the steric hindrance (sterically stabilized dispersions). The former mechanism slows down the rate of aggregation of the nanoparticles due to electrostatic repulsion forces [219], whereas the latter involves grafting of polymer coating due to the addition of polymers acting as steric stabilizers (e.g., polyvinyl pyrrolidone, polysorbate 80, polyethylene glycol, and many more) on the surface of the dispersed nanoparticles inducing thermodynamic stability [220,221]. However, the surface charge of the ZnO nanomaterial suspensions also decide for the eco- and cytotoxicity of these nanomaterials [222]. The nano-ZnO particle dispersion bearing positive charge at cell physiological pH exhibits an enhanced ability to penetrate the cells than the vice versa [223]. 

## 7. Conclusions

The nano-Zn products, particularly the nanoformulations developed for suppression of bacterial, phytoplasma, fungal, or viral diseases in crop plants, can have a gross impact on decreasing the extent of voluminous use of conventional metal(s)-based pesticides. These formulations can be designed for the management of diseases in both open field and closed greenhouse/screen-house conditions and can be applied to crop plants through several application modes. The prior research has shown high effectivity of nano-Zn formulations to curb phytopathogen owing to versatile antimicrobial mechanism of action including photo-oxidation leading to generation of reactive oxygen species, destabilization of the cell membrane, organelles, and other cellular macromolecules, and toxicity due to the release of zinc ions. The zinc nanomaterials have also been utilized for the development of affordable sensor systems for sensitive and early detection of pathogen attack that can be used for predicting the crop losses and for surveillance purposes. Although there are apparent advantages of the use of zinc nanomaterials for diverse benefits, however, their proficient use is limited due to rising concerns about ecohealth deterring aspects of nanomaterials and the bio-/econanotoxicity issues that need to be addressed. The problems such as bioaccumulation across the food chain and food web, complexities of events and components of the plant-soil-atmosphere-pathogen continuum, photo-oxidation properties, and the unprecedented fate of applied nanomaterials in the environment depreciate, comprise, or even negate the advantages of zinc nanomaterials as novel plant disease suppression or eradication agents. Carefully designed protocols and assays dissecting the dimensions and role of nanoscale particles/materials on pathogen and plant can improve our know-how and may direct novel paradigms for adaptation and application of zinc nanomaterials to overt the global food production challenges posed by phytopathogens.

## Figures and Tables

**Figure 1 jof-06-00222-f001:**
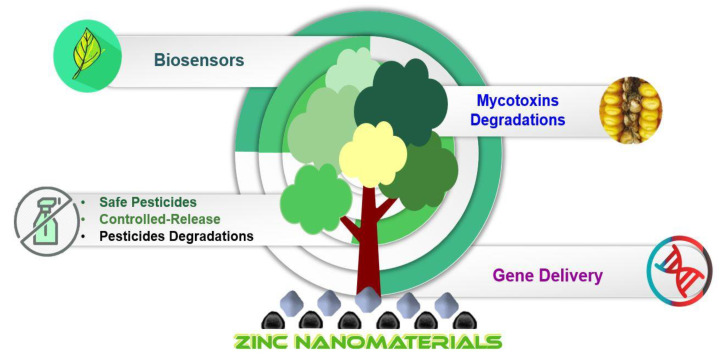
Zinc-based nanomaterials applications in plant pathology.

**Figure 2 jof-06-00222-f002:**
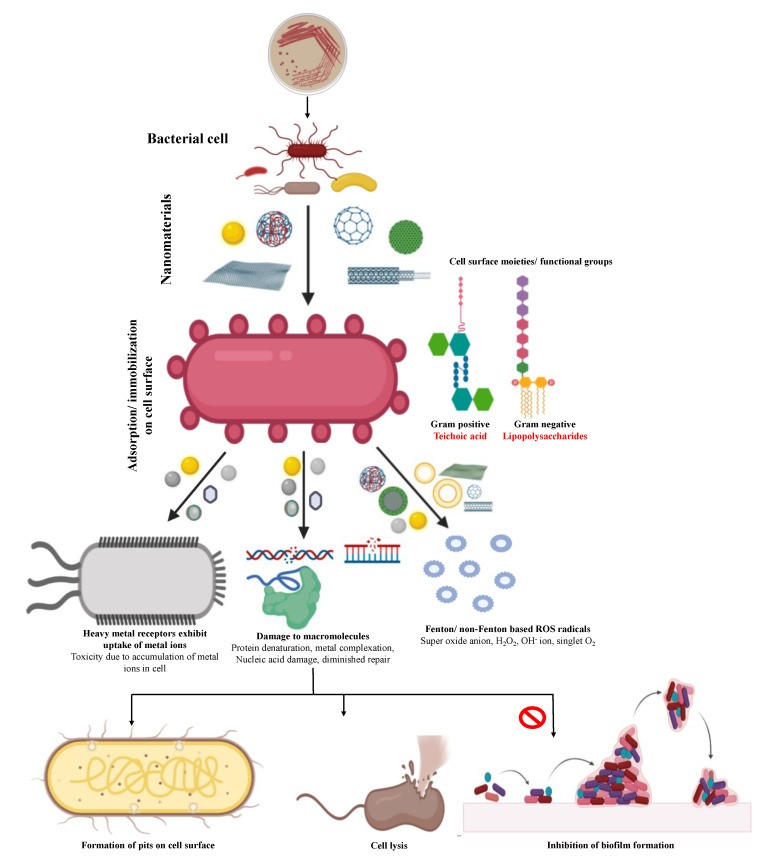
Mechanisms governing the antibacterial potential of different types of nanomaterials.

**Figure 3 jof-06-00222-f003:**
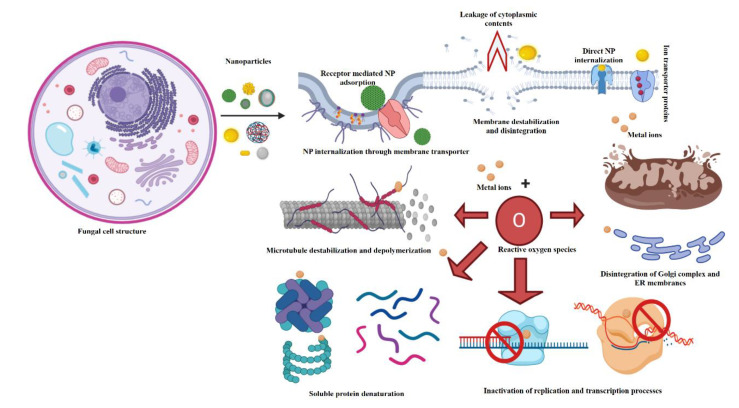
Effect of application of different types of nanoparticles on cellular components and organelles in a fungal cell.

**Figure 4 jof-06-00222-f004:**
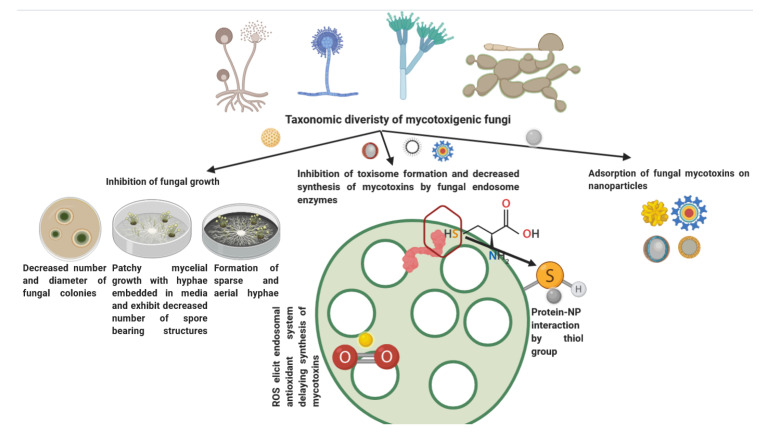
Zinc nanomaterials can exhibit a threefold impact on the production and neutralization of mycotoxins produced by mycotoxigenic fungi.

**Table 1 jof-06-00222-t001:** Antimicrobial potential of zinc nanomaterials on plant pathogenic microbes.

Type of Zn-Nanomaterial Used	Zn-Nanomaterial Characterization	Working Concentration	Study Conditions (Exposure Technique)	Zn-Nanomaterial Application Method	Pathogen Inoculation Technique	Pathogen Studied	Impact	References
**Bacterial pathogens**								
Zinkicide SG4, Zinkicide SG6	2-D nanoplate-like structure (dimensions: 0.2–0.5 mm, thickness: ~10.0 nm) nanoparticulate (size: 4–6 nm)	2000 to 1.96 mg/mL	In vitro assay (broth microdilution technique)	Addition in broth at different working concentrations	Broth inoculation	*X. alfalfae* subsp. *citrumelonis*	Two-fold and 7/8-fold lower MIC for Zinkicide SG4 and SG6, respectively	[116]
ZnO NPs	Commercial formulation (size <100 nm)	0.1 mg mL^−1^	In planta assay	Foliar spray of ZnO NPs suspension (10 mL per lentil plant) under pot culture conditions	Nutrient broth culture (10 mL of 1.2 × 10^5^ CFU mL^−1^) added around the seedling	*Xanthomonas axonopodis* pv. *phaseoli*	Reduction in disease severity on pathogen challenge	[118]
Zinkicide SG4, Zinkicide SG6	2-D nanoplate-like structure (dimensions: 0.2–0.5 mm, thickness: ~10.0 nm) nanoparticulate (size: 4–6 nm)	Zn (30% *w/v*)	In planta assay	-Foliar spray of Zn formulation (10 mL per grapefruit seedling) using air-brush in greenhouse assay -Foliar spray of Zn formulations (3.0 L per grapefruit tree) with a handgun sprayer	Broth culture (10^4^ CFU mL^−1^) in PBS injection-infiltrated in midrib of leaf 3 each site at both surfaces	*Xanthomonas citri* subsp. *citri*	-Reduction in citrus canker disease -Effective disease control comparable or better than Cu_2_O/Cu_2_O-ZnO bactericides (no phytotoxicity)	[116]
ZnONPs	TEM: 41–51 nm	4, 8, and 16 μg mL^−1^	In vitro assay	Variable concentrations of ZnO NPs (10 μL each) dropped on 1-day old bacterial lawn culture	Lawn growth obtained by spread plating of (100 μL, 10^8^ cfu mL^−1^) broth culture followed by incubation for 24 h	*Xanthomonas oryzae* pv. *oryzae* (strain GZ 0003)	Effective antimicrobial agent for bacterial leaf blight of rice	[81]
Cu-Zn hybrid NPs	TEM: 40–100 nm	1000, 500, 200, and 100 µg mL^−1^	In vitro assay	NP formulations added to broth at different concentrations	Broth culture (20 µL, 10^5^ CFU mL^−1^)	*Xanthomonas perforans* (Cu-tolerant GEV485)	Complete inhibition of growth till 24 h of incubation	[117]
Cu-Zn hybrid NPs	TEM: 40–100 nm	500, 200, 100, and 50 µg mL^−1^	In planta assay	Foliar spray on 4-week old seedlings of tomato variety FL 47 under growth chamber conditions	Pathogen inoculation-foliar spray	*Xanthomonas perforans* (Cu-tolerant GEV485)	Statistically highest decrease in disease symptoms at 500 µg/mL	[117]
**Fungal pathogens**								
ZnO NPs	Commercial formulation (< 50 nm particles size)	0, 1, 10, 100, 500, and 1000 μg/mL	In vitro assay (poison food technique)	Supplementation of PDA with different working concentrations	Mycelial plug (5 mm) cut from master culture PDA plate (4-day old growth from edge)	*Alternaria alternata*	-Mean inhibition rate (EC_50_) range 235 and 848 μg/mL -higher efficacy compared to ZnSO_4_	[59]
ZnO NPs/CS-Zn-CuNPs	DLS: 1.5–20 nm TEM: 6–21 nm	0, 30, 60, and 90 µg mL^−1^	In vitro assay (poison food technique)	Addition various working concentrations of prepared nanomaterials in PDA media	Mycelial plug (5 mm) cut from edge of 1-week old fungal growth on PDA media	*Alternaria alternata, B. cinerea, R. solani*	-Highest mycelial inhibition by chitosan mixed Zn-Cu nanocomposite	[62]
3D flower-shaped nanostructured ZnO	FE-SEM: 700–800 nm XRD: crystallite size—42.0 ± 0.8 nm	0.3125–5.0 mM	In vitro assay (broth culture experiment)	Supplementation of broth with different concentrations of Zn nanomaterial	Aqueous conidial suspension (125 µL, 4 × 10^6^ spores mL^−1^) added to Sabouraud dextrose broth (100 mL)	*Aspergillus flavus* Link (UNIGRAS-1231)	-For 1.25–5.0 mM concentrations -78.0% decrease in mycelial growth -99.7% decrease in aflatoxin synthesis	[167]
Metallic (Au/Ag) and ZnO NPs	Commercial formulation DLS: 7 and 477 nm, respectively	50:10 μg/mL	In vitro assay (A. broth microtiter plate test, B. Kirby-Bauer disk diffusion technique)	A. NP suspension (20 μL in 75 μL SDB) B. NP impregnated on sterilized filter paper disks (6 mm diameter)	A. Spore suspension (5 μL, 1 × 10^5^ spores/well) B. Spread plating of spore suspension	*Aspergillus flavus* (NRRL 3518)*/A. fumigatus* (ATCC 1022)	-combination of mix metallic NPs and ZnO-NPs effectively inhibited the fungal growth	[51]
ZNPs	DLS: 30–40 nm TEM: 15–20 nm (average particle size)	50, 100, 250, and 500 ppm	In vitro assay (poison food technique)	Different ZnO NPs concentrations mixed in sterilized PDA media	Fungal spore suspension (3 µL, ~10^4^ mL^−1^) spot plated in center of PDA media plate	*Aspergillus niger*	-dose-dependent decrease in radial growth diameter	[143]
ZnO NPs	Commercial formulation (TEM: 70 ± 15 nm)	0, 3, 6, and 12 mM L^−1^	In vitro assay (poison food technique)	ZnO NPs mixed in different concentrations in PDA media	Aqueous spore suspension (~10^4^ mL^−1^)	*Aspergillus niger* (MTCC-10180), *Fusarium oxysporum* (NCIM-1043, NCIM-1072)	-Significant inhibition in hyphal growth at concentration of 3 mM L^−1^	[144]
ZnO NPs	Leaf extract of derived NPs	200, 300 and 400 µg mL^−1^	In vitro assay (poison food technique)	Supplementation of PDA with different working concentrations of NPs	Fungal disc (5 mm diameter) from 5-day old culture growth	*Alternaria alternata, Botrytis cinerea*	-Concentration-dependent decrease in fungal growth	[145]
A. ZnO NPs, B. ZnO:MgO NPs C. ZnO:Mg(OH)_2_ composite	A. TEM: 22–37 nm B. TEM: 23–30 nm C. TEM: 23–49 nm	Serial dilution ranging from 5 to 0.002 mg mL^−1^	In vitro assay (broth microdilution and agar-media based poison food technique)	DMSO dissolved NPs were diluted with PDB in a geometric progression	Aqueous spore suspension (1 × 10^6^ conidia mL^−1^) added in PDB	*Colletotrichum gloeosporioides*	-ZnO NPs alone exhibited highest inhibition of the hyphal growth -Addition of MgO diminished the antifungal potential of ZnO NPs	[56]
ZnO NPs	TEM: 20 nm (spherical), 37 nm (acicular)	3, 6, 9, and 12 mM L^-1^	In vitro assay (poison food technique)	Supplementation of PDA with different working concentrations of NPs	Mycelial plug (1.5 cm diameter) from 16-day old fungal culture	*Erythricium salmonicolor*	-substantial mycelial growth inhibition at 6 mmol L^−1^	[68]
ZnO NPs	Commercial formulation (size <100 nm)	0, 100, 250, and 500 mg [Zn] L^−1^	In vitro assay (poison food technique)	Different concentrations of ZnO NPs supplemented in mung bean agar media	Mycelial plugs (~0.5 × 1.0 cm) cut from the margins of the 5-day old fungal growth	*Fusarium graminearum*	-dose-dependent inhibition of fungal growth	[146]
ZnO NPs	TEM: 30–40 nm SEM: triangular- to hexagonal-shaped particles XRD: crystallite size—35.69 nm	25, 50, 75, 100, 125, and 140 µg mL^−1^	In vitro assay (broth culture experiment)	Different concentrations of ZnO NPs supplemented in Czapek Dox broth	Spore suspension (10 µL, 10^6^ spores mL^−1^ in peptone water + 0.01% Tween 80) in Czapek Dox broth (100 mL)	*Fusarium graminearum*	In dose-dependent manner -ROS accumulation in treated mycelial -reduction in deoxynivalenol and zearalenone production	[170]
ZnO NPs	TEM: spherical-shaped 30 nm size NPs XRD: wurtzite crystal nature	10, 25, 50, and 100 mM	In vitro assay (poison food technique)	-Variable concentrations added to PDA -Highest Zn-compounds concentration added to PDA	Mycelial disc (6 mm) obtained from 7-day-old fungal cultures from edge	*Fusarium graminearum, Aspergillus flavus, Penicillium citrinum*	-concentration-dependent decrease in hyphal growth -significant decrease in deoxynivalenol and aflatoxin B1 only by ZnO NPs compared to control	[173]
ZnO NPs	DLS: 111.53 ± 1.3 nm TEM: < 100 nm ζ-potential: −15.89 mV	100–800 ppm	In vitro assay (poison food technique)	-Different concentrations of ZnO NPs added to Czapek Dox agar	Mycelial disc (5 mm diameter) was cut from 5-day old culture	*Fusarium moniliforme*	-Less hyphal growth inhibition due larger sized particles	[40]
ZnO NPs	Commercial formulation (size: 70 ± 15 nm)	0, 2, 4, 6, 8, and 12 mg L^−1^	In vitro assay (poison food technique)	Different concentrations of ZnO NPs with autoclaved PD agar medium	Fungal mycelia plug (1 cm diameter) taken from the edge of one-week old culture	*Fusarium oxysporum*	-19.3–77.5% hyphal growth inhibition corresponding to for 2–12 mg L^−1^ ZnO NP concentration	[66]
ZnO NPs	Commercial formulation (spherical-shaped 20–30 ± 10 nm NPs)	25, 50, and 100 ppm	In vitro assay (poison food technique)	Working concentrations of ZnO NPs derived from 1000 ppm stock solution added to sterilized PDA medium	Fungal disc (0.5 cm diameter) obtained from 7-old culture	*Fusarium oxysporum f. sp. betae*	-49.3% inhibition of radial hyphal growth at 100 ppm	[147]
ZnO NPs	Commercial formulation (size: <50 nm)	0–15 mM equivalent to 0–1221 ppm	In vitro assay (automated turbidimetric assay)	ZnO NPs suspension-soaked filter papers	Spore suspension (1.73 × 10^3^ conidia mL^−1^) were serially diluted	*Penicillium expansum*	-MIC: 9.8 mM (798 ppm) and NIC: 1.8 mM (147 ppm)	[148]
A. Zn NPs B. ZnO NPs	A. TEM: mean diameter 264 nm; hydrodynamic diameter: 615.8 nm; ζ-potential: −1.6 ± 3.7 B. TEM: mean particle diameter 19.3 nm; hydrodynamic diameter: 453.3; ζ-potential: 23.3 ± 5.0	0–65 mg L^−1^	In vitro spore germination and infectivity tests	Different concentrations of nano-Zn formulations incubated with fungal spore suspension	Spore suspension (10^6^ spores mL^−1^) mixed with DI	*Peronospora tabacina*	-Inhibition of spore germination frequency spore by Zn NPs, ZnO NPs, and ZnCl_2_ (<10 mg L^−1^) -Significantly higher inhibition by ZnO NPs compared to bulk ZnO -Reduction in leaf infection in tobacco leaf assay	[141]
ZnO and CuO NPs	Commercial formulation	50, 100, 250, and 500 mg L^−1^	In vitro assay (poison food technique)	Different concentrations of NPs amended in autoclaved PDA media	Fungal growth plug (0.5 cm^2^) placed in center of PDA media	*Pythium ultimum, Pythium aphanidermatum*	-Inhibition of growth at low concentrations -morphological changes in the hyphae	[142]
**Viral pathogens**								
ZnO NPs	TEM: 18 nm spherical-shaped particles	A. 100 μg mL^−1^ B. 100 μg mL^−1^ (5 mL NP solution foliar spray for 3, 7, and 12 days)	A. In vitro assay B. In planta assay (*Nicotiana benthamiana*)	A. ZnO NP suspension mixed with purified TMV particles B. Foliar spray of NPs suspensions	A. Purified TMV particles mixed with NPs B. Inoculation by rubbing infected leaves onto the oldest leaf	Tobacco mosaic virus	A. aggregation or breakage of tobacco mosaic virus particles B. marked suppression (35.33%) of TMV invasion in the inoculated leaves	[45]

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
