# Peer review of "Zinc-Based Nanomaterials for Diagnosis and Management of Plant Diseases: Ecological Safety and Future Prospects"

_jof, 2020, doi:10.3390/jof6040222_

Round 1
Reviewer 1 Report
General comments: The review deals with the effects of Zn-based nanomaterials in plant pathogen suppression, analyzing in vitro and in vivo approaches and evidences available on present literature. The manuscript resulted robust and well-constructed. Few suggestions for the authors are included. Minor revisions are suggested.
Paragraph 5. Authors should consider between the molecular tools also available biomarkers of exposure/effects, specifically related to ZnO NPs response, both in case of presence and absence of pathogen infection. There are few examples in the present literature, which include ZnO NPs.
Paragraph 6. Authors should include also a comment related to the sensitive parameters that need to be considered in “improving” nanoformulations in terms of sustainability and safety for environment and human health, analyzing critically each aspect. Either in this case several perspective papers are available in the present literature.
Author Response
Reply to the Reviewer’s comments
- Reviewer 1:
Comments and Suggestions for Authors
General comments: The review deals with the effects of Zn-based nanomaterials in plant pathogen suppression, analyzing in vitro and in vivo approaches and evidences available on present literature. The manuscript resulted robust and well-constructed. Few suggestions for the authors are included. Minor revisions are suggested.
Answer: We graciously thank the learned reviewer for the positive comments. We also thank the reviewer for providing the suggestions which have been incorporated in the revised manuscript.
Paragraph 5. Authors should consider between the molecular tools also available biomarkers of exposure/effects, specifically related to ZnO NPs response, both in case of presence and absence of pathogen infection. There are few examples in the present literature, which include ZnO NPs.
Answer: A paragraph (line number: 445-454) has been incorporated in section 5.0 as indicated by the reviewer.
Paragraph 6. Authors should include also a comment related to the sensitive parameters that need to be considered in “improving” nanoformulations in terms of sustainability and safety for environment and human health, analyzing critically each aspect. Either in this case several perspective papers are available in the present literature.
Answer: A paragraph (line number: 524-543) describing the development of environmentally safe and bio-safe formulations of zinc nanomaterials has been incorporated.
Reviewer 2 Report
Review comments on “Zinc-Based Nanomaterials for Diagnosis and Management of Plant Diseases: Ecological Safety and Future Prospects”
General comment
This is a good review paper about the application of Zn-based nanoparticles application in agriculture. Authors list many cases and examples on the application of Zn/ZnO nanoparticles to control soil microorganisms in current years. And the authors discussed the advantages of Zn-based nanoparticles to some soil microorganisms. But, the first question, I think the topic is not fit for the journal scope, the review topic is focused more on the nanoparticles and little on the pathogen. It is a big problem for acceptance. Second, authors should write more information on how nanoparticles reduce the microorganism activity. And the final one, the authors should list more aspects and reasons that the potential application of nanoparticles in agriculture in the future.
Comments:
- In the Introduction part, the authors write the application of Zn-based nanoparticles to the soil microorganism, but the second part, the authors discuss the Status of use of nanomaterials in plant pathology, I think the Status of use of nanomaterials in plant pathology should come first in the Introduction part, and then you can introduce the Zn-based nanoparticles to the soil microorganism, this order is reasonable;
- The definition of Zn nanoparticles is pesticides or antimicrobial kits? The author should declare it clearly, it is pesticides at the beginning, then it becomes the antimicrobial kits.
- Bacteria is different from the fungal and other microorganisms, the author used the bacteria model to describe the nanoparticles benefits to destroy the bacteria, is that possible for the fungal?
- In this research, the researches list many references of the nanoparticles controlling the microorganism, but nanoparticles have many different properties, authors should list the references with metal/metal oxide nanoparticles.
- Authors need to redesign Table 1. , make the information more clear.
- I think the part “Eco-safety issues of nano-Zinc-derived products and devices” is not necessary with the main part, I think you can raise some ideas about the potential application of Zn-based nanoparticles and future use, such as you also can borrow some aspects about the foliar and root application to improve crop resistance in agriculture use.
- Focus more on fungal, if not, it is not enough for the journal, authors mixed the fungal, bacteria, and microorganism.
Author Response
Reply to the Reviewer’s comments
- Reviewer 2:
General comment
This is a good review paper about the application of Zn-based nanoparticles application in agriculture. Authors list many cases and examples on the application of Zn/ZnO nanoparticles to control soil microorganisms in current years. And the authors discussed the advantages of Zn-based nanoparticles to some soil microorganisms. But, the first question, I think the topic is not fit for the journal scope, the review topic is focused more on the nanoparticles and little on the pathogen. It is a big problem for acceptance. Second, authors should write more information on how nanoparticles reduce the microorganism activity. And the final one, the authors should list more aspects and reasons that the potential application of nanoparticles in agriculture in the future.
Answer: We thank the learned reviewer for appreciating the manuscript. Regarding the three observations, the first thing is that the aim of the review was to present nanomaterials as novel antimicrobials for management of phytopathogens. There is very scanty compiled information on the antimicrobial potential of nanomaterials particularly the zinc nanomaterials for phytopathogen control. We have compiled the available information for all the major phytopathogens with specific emphasis on fungal pathogens.
Replying to the second observation of the reviewer, various mechanisms governing the antimicrobial action of different nanomaterials particularly on zinc nanomaterials has already been described in detail in section 2.1.1. Now this section has been further strengthened with the addition of three sub-sections on bacteria (sub-section 2.1.1.1), fungi (sub-section 2.1.1.2) and viruses (subsection 2.1.1.3). In this section, the variation in the mechanism(s) of action governing antimicrobial potential among the three groups of plant pathogens has been described in detail. Further, the variation in the action mechanisms have been graphically illustrated through three figures (Figure 2 to 4).
For the third observation, we would like to draw attention towards the section 5.0. In this section the use of zinc-based nanomaterials for the fabrication of sensors systems or devices has been described. This application has environmental prudence considering non-contact use of Zn-based nanomaterials besides the benefit of easy, rapid and many-a-times on-farm detection of the plant pathogens. However, as desired by the reviewer, a new section covering potential applications of Zn-based nanomaterials and future use has been incorporated.
Comments:
- In the Introduction part, the authors write the application of Zn-based nanoparticles to the soil microorganism, but the second part, the authors discuss the Status of use of nanomaterials in plant pathology, I think the Status of use of nanomaterials in plant pathology should come first in the Introduction part, and then you can introduce the Zn-based nanoparticles to the soil microorganism, this order is reasonable.
Answer: The paragraph from line number 43 to 51 has been modified and moved to line number 86 to 93 as per the suggestion.
- The definition of Zn nanoparticles is pesticides or antimicrobial kits? The author should declare it clearly, it is pesticides at the beginning, then it becomes the antimicrobial kits.
Answer: Zinc element or its oxidizing compounds have been popularly utilized as pesticides (antimicrobial action for use as bactericide, fungicide, and algaecide) besides the rodenticide, and weedicide applications as described in the section 1.1 of the revised manuscript. Zinc nanoparticles are the novel nano-scale dispersions/ suspensions that are now being evaluated for potential antimicrobial, antioxidant, photocatalytic, pollutant remediation activities. As the review aims for antimicrobial potentials of the zinc nanomaterials for the management of phytopathogens only, in the introduction a broad use of zinc nanomaterials is given followed by its specific role as antimicrobial agent has been highlighted. Some requisite changes have been incorporated to further clarify the objective of the review.
- Bacteria is different from the fungal and other microorganisms, the author used the bacteria model to describe the nanoparticles benefits to destroy the bacteria, is that possible for the fungal?
Answer: We agree to the reviewer’s comment that the bacterial system is different from the eukaryotic fungal cell system. Under the mechanism of action of the various nanomaterials as antimicrobial agents, we have been discussed the prokaryotic bacteria (line number: 157-171, Figure 2), eukaryotic fungi (line number: 172-187, Figure 3) and acellular viruses (line number: 193-229). Here in this section, the basic aim was to segregate the predominant mechanisms governing antimicrobial activity for prokaryotic bacteria vs. the eukaryotic fungal cells and the non-living entities i.e. viruses. The section has been revised and subsections have been incorporated to further clarify the contents.
- In this research, the researches list many references of the nanoparticles controlling the microorganism, but nanoparticles have many different properties, authors should list the references with metal/metal oxide nanoparticles.
Answer: Yes, we agree to the reviewer’s comment that nanomaterials exhibit different functional properties. We would like to clarify here that our compilation was on nano-antimicrobial agents against phyto-pathogens. Further, there are only two specific references on carbon nanomaterial-based composites (reference number 150 and 162) otherwise all the other references include the anti-microbial potential of metal/ metal oxide/, metalloid or non-metal oxide nanoparticles or their composites against the plant pathogens.
- Authors need to redesign Table 1., make the information more clear.
Answer: The Table 1 has been redesigned. New sub-columns have been incorporated to segregate the pathogen inoculation and Zn-nanomaterial application techniques for improved clarity.
- I think the part “Eco-safety issues of nano-Zinc-derived products and devices” is not necessary with the main part, I think you can raise some ideas about the potential application of Zn-based nanoparticles and future use, such as you also can borrow some aspects about the foliar and root application to improve crop resistance in agriculture use.
Answer: A new section entitled ‘Potential application of Zn-based nanoparticles for future use’ has been prepared. The ecosafety issues have been put as sub-section under this section.
- Focus more on fungal, if not, it is not enough for the journal, authors mixed the fungal, bacteria, and microorganism.
Answer: The primary objective of the review was to present the diversity of antimicrobial potential of nanomaterials including Zn-nanomaterials against various plant pathogens. However, fungi have been most elaborately described (sub-sections 2.1.1.2, line number 172 to 187; section 3.2, line number 286-371; Figures 3 and 4). Even the table 1 in the manuscript has more than 5.0 pages describing the antimycotic potential of the zinc nanomaterials. We have updated and enhanced the content in these sections as desired by the reviewer.
Round 2
Reviewer 2 Report
In the revised manuscript, the authors do follow the comments to improve the paper’s structure. And they also make their review contents closer to the Journal of Fungi. I think the manuscript can be accepted by the Journal of Fungi. It is a good and meaningful work to introduce the Zn-based nanoparticles to control soil microorganisms in agriculture.